# Involvement of MicroRNA-1-FAM83A Axis Dysfunction in the Growth and Motility of Lung Cancer Cells

**DOI:** 10.3390/ijms21228833

**Published:** 2020-11-22

**Authors:** Pei-Jung Liu, Yu-Hsuan Chen, Kuo-Wang Tsai, Hui-Ying Yeah, Chung-Yu Yeh, Ya-Ting Tu, Chih-Yun Yang

**Affiliations:** 1Division of Chest Medicine, Department of Internal Medicine, Kaohsiung Veterans General Hospital, Kaohsiung 81362, Taiwan; pjliu20130201@gmail.com; 2Division of Chest Medicine, Department of Internal Medicine, CHENG HSIN General Hospital, Taipei 11221, Taiwan; anemia0829@gmail.com; 3Department of Research, Taipei Tzu Chi Hospital, Buddhist Tzu Chi Medical Foundation, New Taipei 23124, Taiwan; kwtsai6733@gmail.com (K.-W.T.); new0224@hotmail.com (Y.-T.T.); 4Department of Medical Education and Research, Kaohsiung Veterans General Hospital, Kaohsiung 81362, Taiwan; hyyeh@vghks.gov.tw (H.-Y.Y.); cyyeah@vghks.gov.tw (C.-Y.Y.); 5Division of Chest Medicine, Kaohsiung Municipal Min-Sheng Hospital, Kaohsiung 802213, Taiwan

**Keywords:** lung cancer, FAM83A, miR-1-3p

## Abstract

Lung cancer is the most prevalent types of cancer and the leading cause of cancer-related deaths worldwide. Among all cancers, lung cancer has the highest incidence, accompanied by a high mortality rate at the advanced stage. Favorable prognostic biomarkers can effectively increase the survival rate in lung cancer. Our results revealed *FAM83A* (Family with sequence similarity 83, member A) overexpression in lung cancer tissues compared with adjacent normal tissues. Furthermore, high *FAM83A* expression was closely associated with poor lung cancer survival. Here, through siRNA transfection, we effectively inhibited *FAM83A* expression in the lung cancer cell lines H1355 and A549. FAM83A knockdown significantly suppressed the proliferation, migration, and invasion ability of these cells. Furthermore, *FAM83A* knockdown could suppress Epidermal growth factor receptor (EGFR)/Mitogen-activated protein kinase (MAPK)/Choline kinase alpha (CHKA) signaling activation in A549 and H1355. By using a bioinformatics approach, we found that *FAM83A* overexpression in lung cancer may result from miR-1-3p downregulation. In summary, we identified a novel *miR-1-FAM83A* axis could partially modulate the EGFR/choline phospholipid metabolism signaling pathway, which suppressed lung cancer growth and motility. Our findings provide new insights for the development of lung cancer therapeutics.

## 1. Introduction

Lung cancer is the most common cancer and the leading cause of cancer-related deaths worldwide. According to histological features and appearance of malignant cells, lung cancer is classified into 2 types: non–small-cell lung cancer (NSCLC) and small-cell lung cancer. Of them, NSCLC is more common, accounting for approximately 85% of all lung cancers. More than 40% of NSCLC cases are metastatic at diagnosis [1]. The primary subtypes of NSCLC include lung adenocarcinoma (LUAD), lung squamous cell carcinoma (LUSC), and large-cell carcinoma [2]. Over the past decade, the treatment of NSCLC and development of therapeutic approaches have improved considerably [3]. However, the mortality rate of patients with advanced lung cancer remains high. The development of reliable diagnostics and identification of prognostic biomarkers are essential to increasing the survival rate of patients with lung cancer.

Lung cancer progression depends on the effects of individual oncogenes; therefore, oncogene-targeted therapy has been confirmed to be effective. However, cancer cells can also be adapted by compensatory activation of the downstream pathway and develop resistance to target therapy. Family with sequence similarity 83, member A (*FAM83A*) is a gene widely expressed in eukaryotic cells, particularly in the epithelial cells of the skin, bladder, kidneys, and lungs [4]. *FAM83A* is highly expressed in several human cancers, including pancreatic, breast, and lung cancer and hepatocellular carcinoma [5,6,7,8,9,10,11]. It plays an oncogenic role in promoting cancer cell proliferation and metastasis through the mitogen-activated protein kinase (MAPK)/extracellular signal-regulated kinase and phosphatidylinositol-3-kinase (PI3K)/protein kinase B (AKT)/mammalian target of rapamycin pathways [9,12,13,14,15,16,17]. However, *FAM83A* overexpression mechanism in human cancer and its potential effects on lung cancer have yet to be established. In this study, we assessed the effects of FAM83A on lung cancer and reported for the first time that microRNA participates in *FAM83A* overexpression in lung cancer.

## 2. Results

### 2.1. Identification of Prognostic Biomarkers for Lung Cancer

We identified genes whose dysfunction is involved in lung cancer progression. Data on a total of 511 RNA transcriptomes and the clinical data of patients with LUAD were downloaded from The Cancer Genome Atlas (TCGA) database. Analysis of the correlations between the clinical characteristics and overall survival of patients with LUAD revealed that the advanced stage was associated with poor survival (crude hazard ratio (cHR) 2.76, 95% CI 2.00–3.80, *p* < 0.001; Table 1). We further identified differentially expressed genes between patients with stage I + II and stage III + IV lung cancer. The genes were selected according to the following criteria: FPKM > 1, fold change of >1.5 or <0.75, and *p* < 0.05 (Figure 1). Compared with those with stage I + II lung cancer, patients with stage III + IV lung cancer demonstrated significantly increased expression of 12 candidate genes but significantly decreased expression of 26 candidate genes (Appendix A). As Figure 1B shows, *FAM83A* expression was significantly upregulated in LUAD tissues compared with in adjacent normal tissues. *FAM83A* expression was also significantly increased in patients with advanced lung cancer, large tumor size, and lymph node metastasis (Table 2). As Figure 2 shows, *FAM83A* expression was significantly elevated in patients with advanced lung cancer (*p* = 0.02 and *p* < 0.001 for stage I vs. stages II and III + IV, respectively), lymph node metastasis (*p* = 0.03 and *p* < 0.001 for N0 vs. N1 and >N2, respectively) and large tumor size (*p* < 0.001 and *p* < 0.001 for T1 vs. T2 and T3 + T4, respectively). *FAM83A* expression was also significantly increased in LUSC tissues compared with in adjacent normal tissues (*p* < 0.001) (Appendix A). However, no correlation between *FAM83A* expression and the clinicopathological features of patients with LUSC was noted (Appendix A).

### 2.2. High FAM83A Expression Correlated with Worse Survival in Patients with LUAD

We assessed the relationship between *FAM83A* expression and postoperative survival of patients with lung cancer. Firstly, we defined a cutoff value for *FAM83A* expression levels, which was determined using receiver operating characteristic analysis. Patients with LUAD were separated into two groups representing high and low *FAM83A* expression on the basis of this cutoff value (30.4). The Kaplan-Meier survival curves showed that compared with low levels, high expression levels of *FAM83A* were closely associated with shorter survival duration (Figure 2D). Univariate Cox regression analysis revealed that high expression of *FAM83A* was correlated with poor survival (cHR 2.57, 95% CI 1.87–3.54, *p* < 0.001; Table 3). Multivariate Cox regression analysis found high *FAM83A* expression to be an independent prognostic biomarker for overall survival in LUAD (adjusted hazard ratio (aHR) 2.17, 95% CI 1.57–3.01, *p* < 0.001; Table 3). Based on the defined cutoff values (19.6), a significant correlation between high *FAM83A* expression and poor survival in LUSC was demonstrated in both univariate (cHR 1.52, 95% CI 1.14–2.05, *p* = 0.005) and multivariate analysis (aHR 1.49, 95% CI 1.11–2.00, *p* = 0.008; Appendix A). In summary, *FAM83A* may be an independent prognostic biomarker for overall survival in lung cancer.

### 2.3. FAM83A Regulates Lung Cancer Cell Growth and Motility

To explore the role of *FAM83A* in lung cancer, we assessed its effects on lung cancer cell growth. We knocked down its expression in H1355 and A549 cells through transfection of siRNAs, revealing that expression levels of *FAM83A* could be inhibited by si*FAM83A*#1 or si*FAM83A* #2 transfection (Figure 3A,B). *FAM83A* knockdown substantially suppressed cell proliferation and colony formation ability in both cell lines (Figure 3C–H). The expression levels of cyclin B and D, which are associated with the cell cycle in lung cancer, were reduced whereas that of p27 was increased in the A549 and H1355 cells (Figure 3I,J). *FAM83A* knockdown substantially inhibited the migration and invasion ability of both the A549 and H1355 cells (Figure 4A–F).

### 2.4. Identification of FAM83A-Associated Signaling Pathways

To assess the mechanism of *FAM83A* knockdown–induced inhibition of growth and invasion ability of lung cancer cells, we performed transcriptome profiling using a microarray approach. The biological function of *FAM83A* knockdown is very similar in A549 and H1355 cells. Therefore, we only performed gene expression profiles on A549 cells with *FAM83A* knockdown. From the microarray data, we identified the putative signaling pathways in *FAM83A* that suppress lung cancer growth and motility. We analyzed 4 transcriptome profiles: those of 2 controls and siFAM83A#1 and si*FAM83A*#2 with *FAM83A* knockdown. We analyzed the sample diversity using principal coordinate analysis. As Appendix A shows, the gene expression pattern of the 2 controls were similar, and those of si*FAM83A* #1 and si*FAM83A* #2 were distinct from the control groups. We further identified differentially expressed genes in A549 cells with *FAM83A* knockdown: 335 that were significantly upregulated (>1.5-fold) and 69 that were significantly downregulated (Appendix A). Pathway enrichment analysis revealed that these 404 genes are critically involved in 16 signaling transduction pathways: Alzheimer disease, thermogenesis, choline metabolism in cancer, renin secretion, Parkinson disease, oxidative phosphorylation, gap junction, glutamatergic synapse, cortisol synthesis and secretion, long-term potentiation, degradation of valine, leucine, and isoleucine, endocytosis, complement and coagulation cascades, propanoate metabolism, legionellosis, and the phospholipase D signaling pathway (Appendix A). We deduced that choline metabolism signaling may be involved in FAM83A knockdown–induced inhibition of growth and motility ability of lung cancer cells. Therefore, we examined the Epidermal growth factor receptor (EGFR)/MAPK/Choline kinase alpha (CHKA) signaling activity in A549 and H1355 cells further with *FAM83A* knockdown. As illustrated in Figure 5A,B, the expression levels of EGFR and CHKA, and the phosphorylation of EGFR and MAPK were reduced in lung cancer cells with *FAM83A* knockdown. These results indicated that *FAM83A* knockdown–induced inhibition of lung cancer cell growth and motility was partially caused by EGFR/MAPK/CHKA signaling activity suppression.

### 2.5. miR-1-3p Suppresses FAM83A Expression by Targeting the 3 Prime Untranslated Region of FAM83A

Our data revealed high *FAM83A* expression in LUAD tissues and its contribution to disease progression. However, the mechanisms underlying abnormal *FAM83A* expression remain unclear. MicroRNAs are small nuclear noncoding RNAs that suppress gene translation or promote mRNA degradation by binding to the 3′ untranslated region (3′UTR). We attempted to identify the miRNAs that regulate *FAM83A* expression in lung cancer. The microRNA.org prediction tool selected 13 miRNA candidates expected to bind to the 3′UTR of *FAM83A*. We then examined the expression of these miRNAs in LUAD by analyzing TCGA database (Figure 6A). As miRNAs suppress target gene expression, we expected them to interact directly with the 3′UTR of *FAM83A* and that *FAM83A* expression would be negatively correlated with target genes in LUAD. We found that *miR-1* regulated *FAM83A* expression by targeting its 3′UTR. Moreover, *miR-1* expression was both negatively correlated with *FAM83A* expression and significantly reduced in LUAD (Figure 6B,C). Low miR-1-3p expression was significantly correlated with advanced stage and worse overall survival in LUAD (Figure 6D and Table 4 and Table 5).

Furthermore, both *FAM83A* mRNA and protein expression levels in lung cancer cells was considerably reduced after *miR-1* mimic transfection (Figure 7A–C). We cloned the 3′UTR sequence into the pmiR-REPORT Luciferase miRNA Expression Reporter Vector System. As Figure 7D,E shows, luciferase activity was significantly reduced in A549 cells after miR-1-3p cotransfection. When the miR-1-3p binding sites were mutated, this suppression was eliminated (Figure 7F), indicating that miR-1-3p inhibited *FAM83A* expression by directly binding to its 3′UTR. To assess the effects of miR-1-3p-*FAM83A* axis activity on LUAD, we classified the combined expression of *FAM83A* and miR-1-3p into 3 groups: high *FAM83A* and low miR-1-3p, low or high levels of both *FAM83A* and miR-1-3p, and low *FAM83A* and high miR-1-3p. As Figure 7G shows, high *FAM83A* and low miR-1-3p expression were significantly associated with poor survival in patients with LUAD.

## 3. Discussion

Evidence of frequent overexpression of *FAM83A* in human cancers, including lung and breast cancer and pancreatic ductal adenocarcinoma, is increasing [5,6,9,11,18]. In a study by Shi et al., TCGA database analysis of LUAD and LUSC demonstrated a positive correlation between *FAM83A* and its antisense RNA *FAM83A-AS1* [8]. As *FAM83A-AS1* is located at the intron of *FAM83A*, both *FAM83A* and *FAM83A-AS1* may be regulated through chromatin remodeling. To our best knowledge, until now, no studies have described the mechanism of *FAM83A* overexpression in human cancers. In a study by Li et al., bioinformatics analysis found increased *FAM83A* expression in patients with lung cancer who smoked compared with in those who did not, suggesting that cigarette smoking can induce *FAM83A* expression [19].

In this study, we reported for the first time that low miR-1-3p expression may contribute to *FAM83A* overexpression in human lung cancer. miR-1-3p was significantly downregulated in lung cancer tissues compared with in adjacent normal tissues. Previous study indicated that *miR-1-3p* is a conserved miRNA with high expression in the muscle tissues, but low abundant in normal lung tissues [20]. In present study, miR-1-3p was indicated to have higher expression in the corresponding adjacent normal tissue than LUAD by analyzing TCGA database. These results implied that *miR-1-3p* expression may be gradually induced when the lung tissue begins to develop precancerous lesions. However, the expression levels of *miR-1-3p* might be further reduced in a malignancy LUAD. This hypothesis needs more experiments to demonstrate it in the future. A study reported that ectopic expression of miR-1-3p in lung cancer cells suppresses proliferation, impairs cell cycle progression, and inhibits migration and invasion ability by silencing *ANXA2* expression [21]. Jiao et al. demonstrated that miR-1-3p sensitizes hepatocyte growth factor–induced gefitinib-resistant lung cancer cells by modulating c-Met signaling [22]. These findings indicate that miR-1-3p suppresses tumor growth by regulating the growth and motility of lung cancer cells. *FAM83A* is a novel target gene for miR-1-3p in lung cancer, and *FAM83A* overexpression may result from downregulation of miR-1-3p.

Symptom nonspecificity and early-stage metastasis make early diagnosis of lung cancer difficult. To establish a prognostic risk model, therefore, numerous studies have investigated prognostic factors for lung cancer. Several recent studies reported that *FAM83A* was significantly overexpressed in lung cancer tissues compared with in adjacent normal tissues, and that this high expression were closely associated with poor prognosis of patients with lung cancer [7]. Zhang et al. studied the expression of 4 genes, *MYO1E*, *ERO1L*, *C1QTNF6*, and *FAM83A*, to build a prognostic panel for lung cancer. Functional enrichment analysis indicated that dysfunction in these genes are involved in regulation and progression of the cell cycle, synthesis and assembly of nucleic acids, and histone modification [23]. Lee et al. suggested a positive correlation of *FAM83A* with therapeutic resistance of tyrosine kinase inhibitors in breast cancer. *FAM83A* interacts with and causes phosphorylation of proto-oncogene c-RAF and PI3K p85 upstream of MAPK and downstream of the epidermal growth factor receptor [9]. The present study demonstrates that *FAM83A* expression may be a prognostic biomarker for pathological stage, lymph node metastasis, and overall survival of patients with lung cancer.

*FAM83A* overexpression promotes cell proliferation and metastasis in lung cancer through the modulation of cancer-related signaling pathways, including PI3K/ATK/Snail, Wnt, Hippo, and MAPK [24,25,26,27]. Studies have reported that choline kinase is overexpressed in lung cancer and has been demonstrated to play a critical role in the onset of human cancer [28,29]. We revealed that *FAM83A* knockdown–induced inhibition of lung cancer cell growth and motility was partially caused by the suppression of EGFR/MAPK/CHKA signaling activity. Jiao et al. reported that miR-1-3p expression reduced the phosphorylation of EGFR in lung cancer cells [22]. Similar results were observed for neck squamous carcinoma cells by using *miR-1* to mimic transfection [30]. Furthermore, in silico analysis revealed that miR-1-3p may be targeted to the 3′UTR of *FAM83A*. These findings suggest that the *miR-1-FAM83A* axis is critical in lung cancer cell growth and metastasis.

In summary, our findings demonstrated that *FAM83A* plays an oncogenic role in regulating lung cancer cell growth and motility via modulating EGFR/MAPK/CHKA signaling activity. In addition, we reported for the first time that miR-1-3p dysfunction may contribute to *FAM83A* overexpression and that high *FAM83A* expression could be a biomarker for poor lung cancer prognosis. Our findings can serve as a reference for biomarker identification or therapeutic development for LUAD.

## 4. Materials and Methods 

### 4.1. Cell Culture

Lung cancer cell lines A549 and H1299 were obtained from the American Type Culture Collection and sustained in Roswell Park Memorial Institute Medium 1640 supplemented with 10% inactivated fetal bovine serum (Invitrogen, Carlsbad, CA, USA). Total RNA was isolated using TRIzol reagent (Invitrogen, Carlsbad, CA, USA) according to the manufacturer’s instructions. The concentrations were measured on a Nanodrop 1000 spectrophotometer (Nanodrop Technologies, LLC, Wilmington, DE, USA).

### 4.2. Gene Expression Profiles According to Cancer Genome Atlas Data

The Cancer Genome Atlas (TCGA) program collects both cancerous and corresponding normal tissues from patients with LUAD and LUSC. We accessed downloaded TCGA data on RNA sequences in LUAD and LUSC tissues from the Genomic Data Commons Data Portal. The clinical information of patients with LUAD and LUSC was also downloaded. The expression profiles of 515 LUAD and 59 corresponding adjacent normal tissue samples as well as 501 LUSC and 49 corresponding adjacent normal tissue samples were obtained from the TCGA data portal. The transcriptome profiles of 485 patients with LUAD and 494 patients with LUSC were used to perform overall survival analysis using the Kaplan–Meier method. The clinical pathological stages were assessed using the eighth edition of the TNM staging system.

### 4.3. Real-Time Reverse Transcription Polymerase Chain Reaction

Total RNA was reverse transcribed using random primers and SuperScript III Reverse Transcriptase according to the manufacturer’s instructions (Invitrogen). Complementary DNA was then subjected to gene expression analysis using Synergy Brands (SYBR) Green Master Mix (Applied Biosystems, Foster City, CA, USA) on the 7900 HT Fast Real-Time PCR System (Applied Biosystems). Glyceraldehyde 3-phosphate dehydrogenase (GAPDH) was used as an internal control. The sequences of all primers used are presented as follows:GAPDH-F: TGCACCACCAACTGCTTAGCGAPDH-R: GGCATGGACTGTGGTCATGAGFAM83A-F: CCCTATAAAGAGTGGCAACAGFAM83A-R: AACAGTGAGCAAACACACCG

### 4.4. RNA Interference Knockdown of FAM83A

H1355 or A549 cells were transfected with si-*FAM83A*#1 and si-*FAM83A*#2 oligonucleotides directed against *FAM83A* (Sigma, Billerica, MA, USA), respectively. Random sequence siRNA oligonucleotides (Sigma) were used as a negative control. After 48 h of transfection, *FAM83A* expression was confirmed through real-time reverse transcription polymerase chain reaction.

### 4.5. Colony Formation Assay

In total, 4000 cells with *FAM83A* knockdown were seeded in 6-well plates and then incubated at 37 °C for 2 weeks. The colonies were fixed with 3.7% formaldehyde for 10 min and stained with crystal violet. Relative colony formation ability was then determined using a spectrophotometer at a wavelength of 620 nm.

### 4.6. Cell Proliferation, Migration, and Invasion Assay

After the si*FAM83A* oligonucleotides were transfected into the lung cancer cells, a total of 5000 cells were seeded in 96-well plates. Cell proliferation was then measured at 0, 1, 2, 3 and 4 days using the CellTiter-Glo One Solution Assay (Promega Corporation, Madison, WI, USA).

### 4.7. Cell Migration and Invasion Ability

The migration and invasion abilities of the lung cancer cells with *FAM83A* knockdown were assessed in vitro in transwell chambers (Costar, Lowell, USA), as we previously described [31]. All experiments were repeated 3 times.

### 4.8. Microarray Data Analysis

The total RNA sources used were as follows: (1) control #1 cells, (2) control #2 cells, (3) si*FAM83A*-#1 cells, and (4) si*FAM83A*-#2 cells. The complementary DNA probes were derived from paired RNA samples from the *FAM83A* knockdown cells or the control cells. The probes were labeled using Cy3-dCTP (green) or Cy5-dCTP (red) and then spotted onto microarray chips. The microarray experiments and data analysis were performed by Welgene Biotech (Taipei, Taiwan) using Agilent Oligo Chips. The differentially expressed genes (genes with *FAM83A* knockdown v.s. control with fold change ≥1.5 and ≤0.75 and *p* < 0.05) were selected from the microarray data. These differentially expressed genes were subjected for gene ontology analysis using Database for Annotation, Visualization and Integrated Discovery (version 6.8) [32] to identify the significantly enriched pathways.

### 4.9. Western Blotting

The total cell lysates were extracted with the radioimmunoprecipitation assay buffer (50 mM Tris HCl, pH 8.0, 150 mM NaCl, 1% NP-40, 0.5% deoxycholic acid, 0.1% sodium dodecyl sulfate). Total proteins were separated through electrophoresis in 6–10% sodium dodecyl sulfate-polyacrylamide gel and transferred onto nitrocellulose filter membranes (Millipore, Billerica, MA, USA). The membranes were then incubated with a blocking buffer for 1 h at room temperature, and incubated with the primary antibodies overnight at 4 °C. For a description of these antibodies, please refer to our previous study [33]. The membranes were then incubated with a horseradish peroxidase–conjugated secondary antibody for 1 h at room temperature to detect the primary antibody. In this study, the primary antibodies were used: CCNB1 (1:1000; 55004-1-AP, Proteintech Group, Inc., Rosemont, IL, USA), CCND1 (1:200; MA5-16356, Thermo Fisher Scientific Inc., Waltham, MA, USA), CDKN1B (p27) (1:500; 25614-1-AP, Proteintech), *FAM83A* (1:500, 20618-1-AP, Proteintech), EGFR (1:1000; #4267, Cell Signaling Technology, Inc., Beverly, MA, USA), p-EGFR (1:1000; #3777, Cell Signaling Technology, Inc.), MAPK (1:1000; #9107, Cell Signaling Technology, Inc.), p-MAPK (1:1000; #4370, Cell Signaling Technology, Inc.), CHKA (1:1000; #13422, Cell Signaling Technology, Inc.), and Actin (ACTB) (1:5000; MAB1501, EMD Millipore, Billerica, MA, USA).

Finally, the proteins were visualized using the WesternBright ECL HRP substrate (Advansta Inc., San Jose, CA, USA) and captured using the BioSpectrum 500 Imaging System (UVP, LLC, Upland, CA, USA).

### 4.10. Stem–Loop Reverse Transcription PCR

According to the manufacturer’s instructions (Invitrogen, Carlsbad, CA, USA) and as we described previously [34], total RNA was reverse transcribed through a stem–loop reverse transcription reaction by using miR-1-3p reverse transcription primers (5′-CTCAACTGGTGTCGTGGAGTCGGCAATTCAGTTGAGATACATAC-3′) and SuperScript III Reverse Transcriptase. Gene expression was assessed using an SYBR Green I assay (Applied Biosystems) and the expression level of miR-1-3p was normalized to that of U6 (ΔCt = miR-1-3p Ct-U6 Ct). The sequence of primers of real-time PCR was as follows:miR-1-3p-GSF: 5′-CGGCGGTGGAATGTAAAGAAGT-3′Universal reverse: 5′-CTGGTGTCGTGGAGTCGGCAATTC-3′U6-F: 5′-CTCGCTTCGGCAGCACA-3′U6-R: 5′-AACGCTTCACGAATTTGCGT-3

### 4.11. miR-1-3p Mimics Trasfection

A549 cells were transfected with 10 nM of miRNA-1-3p mimics (sense: 5′-UGGAAUGUAAAGAAGUAUGUAU-3′; antisense: 5′-ACAUACUUCUUUACAUUCCAUU-3′) or a random siRNA sequence as control (N.C) using Lipofectamine RNAiMAX reagent (13778150, Thermo Fisher Scientific Inc., Waltham, MA, USA).

### 4.12. miRNA Target Candidates and Luciferase Reporter Assay

Identification of miRNA candidates for binding to 3′-UTR of *FAM83A* mRNA was done using the prediction tool on microRNA.org [35]. The 3′-UTR sequences and seed region mutant of *FAM83A* were cloned into a pMIR-REPROT vector (AM5795, Thermo Fisher Scientific). The pMIR-REPROT-*FAM83A* or pMIR-REPROT-*FAM83A*_(mutant)_ vector was cotransfected with or without the miR-1-3p mimics into cells using Lipofectamine 2000 (Invitrogen, Thermo Fisher Scientific). After 24 h of transfection, the luciferase activity was examined using the Dual-Glo Luciferase Assay System (Promega Corporation, Madison, WI, USA). The detailed information was described in our previous study [34]. 

### 4.13. Statistical Analysis

The TCGA data for the expression levels of *FAM83A* or miR-1-3p in patients with LUAD were analyzed using Student *t* tests. Cumulative survival curves were generated using the Kaplan-Meier method, and differences between survival curves were compared using the log-rank test. The correlations between *FAM83A* and miR-1-3p in LUAD tissues were analyzed using Pearson correlation. Cell proliferation, colony formation, migration and invasion experiments were performed in triplicate. The histograms present the mean values, and the error bars indicate the standard deviation. These data were analyzed using Student *t* tests. The difference was considered to be significant when *p* < 0.05.

## Figures and Tables

**Figure 1 ijms-21-08833-f001:**
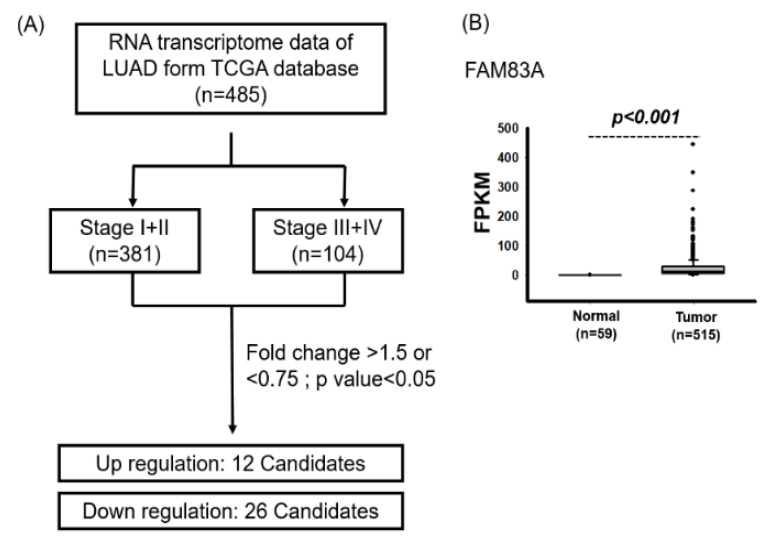
Identification of candidate genes associated with stage in lung adenocarcinoma (LUAD) through TCGA database analysis. (**A**) Flowchart of the identification process. (**B**) Increased *FAM83A* expression in LUAD tissues compared with that in adjacent normal tissues.

**Figure 2 ijms-21-08833-f002:**
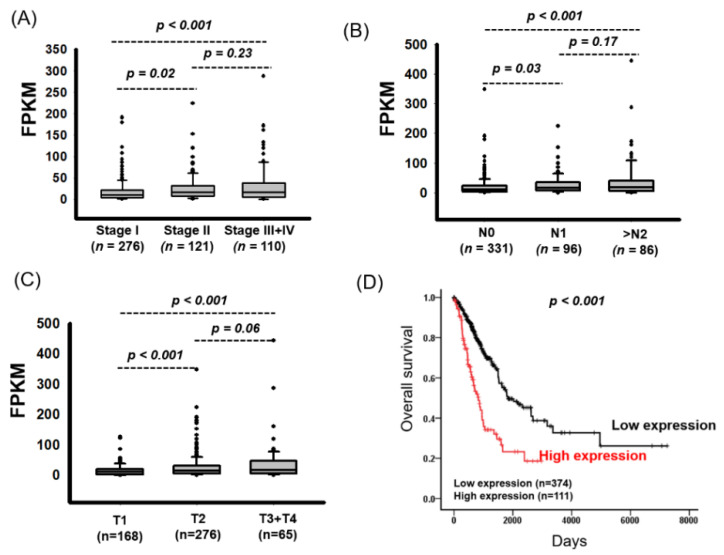
Association of *FAM83A* Expression with Poor LUAD Prognosis. (**A**) *FAM83A* expression in LUAD by pathological stage. (**B**) *FAM83A* expression in LUAD by lymph node metastasis status. (**C**) *FAM83A* expression in LUAD by T stage. (**D**) Kaplan-Meier survival curves showing the effect of *FAM83A* expression on overall survival in LUAD.

**Figure 3 ijms-21-08833-f003:**
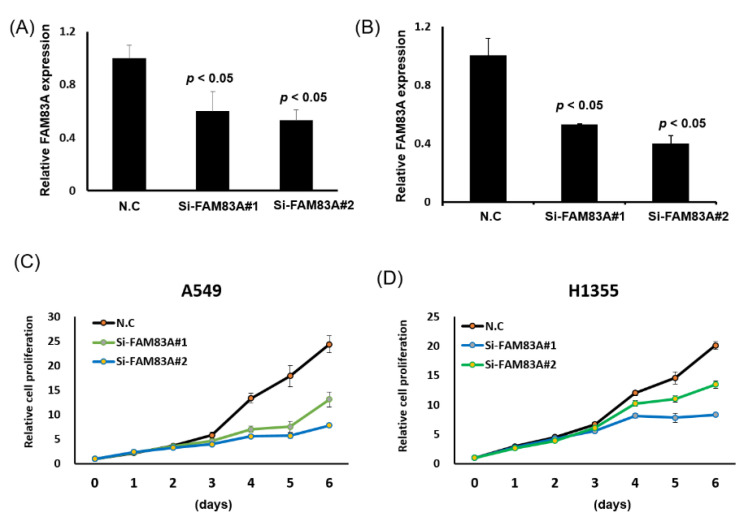
Suppression of lung cancer cell growth and motility by *FAM83A* knockdown. (**A**,**B**) *FAM83A* expression levels in A549 and H1355 cells knocked down using siRNA transfection by real-time reverse transcription polymerase chain reaction. (**C**,**D**) Cell proliferation in A549 and H1355 cells with *FAM83A* knockdown. (**E**–**H**) Quantification of colony formation ability of the A549 and H1355 cells. (**I**,**J**) Western blot analysis of expression levels of cell cycle–related genes in the A549 and H1355 cells. N.C. indicates a random siRNA sequence control. All experiments were performed in triplicate, and the data were analyzed using Student’s *t* test. Differences were considered significant when *p* < 0.05.

**Figure 4 ijms-21-08833-f004:**
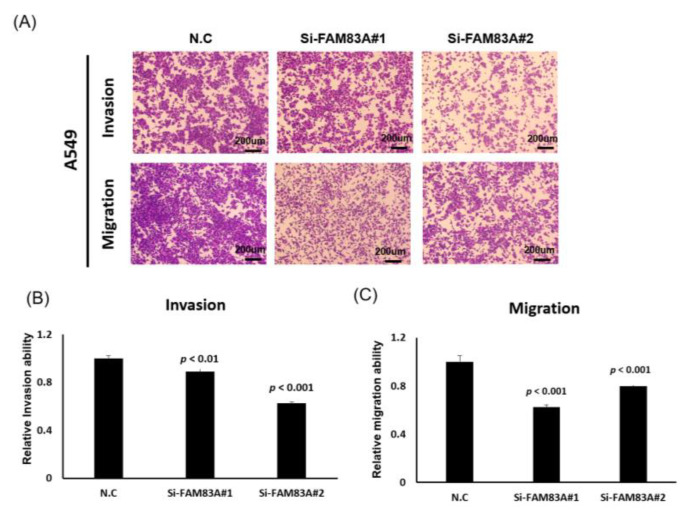
Suppression of lung cancer cell motility by *FAM83A* knockdown. (**A**,**B**) Cell invasion and migration in A549 cells with *FAM83A* knockdown. (**C**) Quantification of invading and migrating A549 cells. (**D**) Cell invasion and migration in H1355 cells with *FAM83A* knockdown. (**E**,**F**) Quantification of invading and migrating H1355 cells. N.C. indicates a random siRNA sequence control. All experiments were performed in triplicate, and the data were analyzed using Student’s *t* test. Differences were considered significant when *p* < 0.05. scale bar = 200 μm.

**Figure 5 ijms-21-08833-f005:**
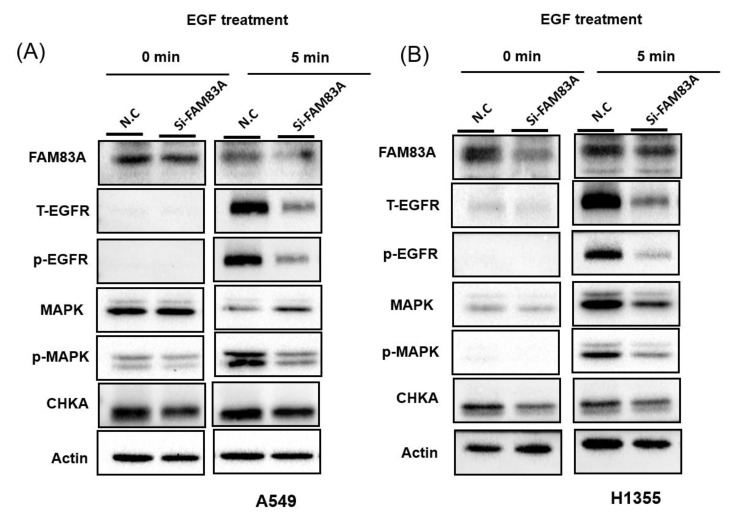
*FAM83A* knockdown–induced inhibition of lung cancer cell growth and motility by choline metabolism signaling. After 5 min of Epidermal growth factor (EGF) treatment, the expression levels of Epidermal growth factor receptor (EGFR), Phosphorylation-EGFR (p-EGFR), Mitogen-activated protein kinae (MAPK), Phosphorylation-MAPK (p-MAPK), and Choline kinase alpha (CHKA) in (**A**) A549 and (**B**) H1355 cells with FAM83A knockdown were assessed using Western blot assay. N.C. indicates a random siRNA sequence control. All experiments were performed in triplicate, and the data were analyzed using Student’s *t* test. Differences were considered significant when *p* < 0.05.

**Figure 6 ijms-21-08833-f006:**
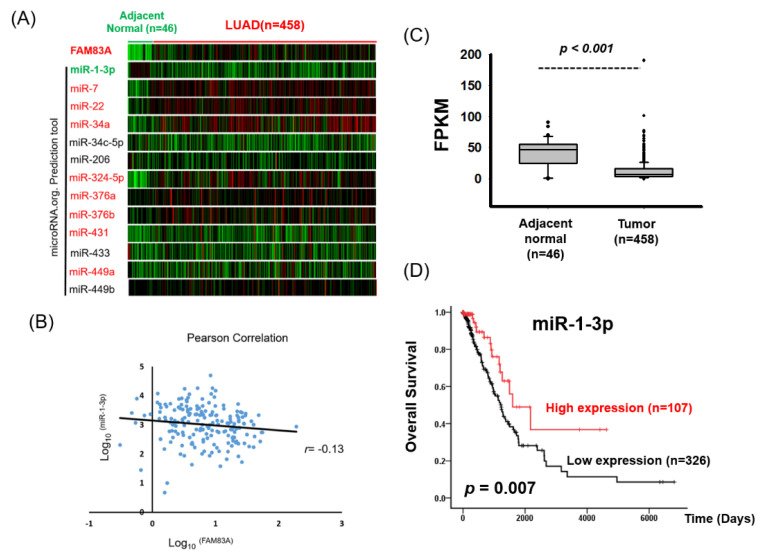
Identification of putative miRNA candidates for *FAM83A* knockdown through a bioinformatics approach. (**A**) Identification of miRNA candidates to regulate *FAM83A* expression. Expression levels of the 13 miRNAs and *FAM83A* in 458 LUAD and 46 normal tissues are shown as heat maps. Candidate genes whose expression was significantly increased (*p* < 0.01) or reduced in lung cancer tissues compared with in adjacent normal tissues are labeled in red and green, respectively. (**B**) Pearson correlation analysis of miR-1-3p and *FAM83A* expression in 185 patients with LUAD. (**C**) TCGA database analysis of miR-1-3p expression in lung cancer. (**D**) Kaplan-Meier survival curves showing the effects of miR-1-3p expression on overall survival of patients with LUAD.

**Figure 7 ijms-21-08833-f007:**
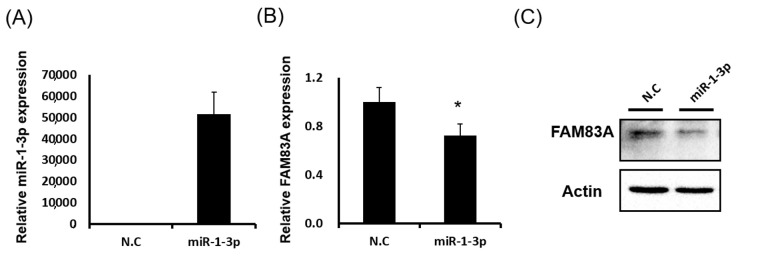
Suppression of *FAM83A* expression by miR-1-3p through direct binding to the 3′ untranslated region (3′UTR) of *FAM83A*. (**A**) miR-1-3p expression in lung cancer cells with miR-1-3p mimic transfection. (**B**) *FAM83A* mRNA expression in lung cancer cells with miR-1-3p and negative control transfection. (**C**) *FAM83A* protein expression in lung cancer cells with miR-1-3p and negative control transfection. (**D**–**F**) Reporter construct with wild-type *FAM83A* 3′UTR or binding site mutant after miR-1-3p cotransfection into lung cancer cells, followed by assessment of luciferase activity. (**G**) Effects of combined *FAM83A* and miR-1-3p expression on overall survival in lung cancer. N.C. indicates a random siRNA sequence control. All experiments were performed in triplicate, and the data were analyzed using Student’s *t* test. Differences were considered significant when * *p* < 0.05.

**Table 1 ijms-21-08833-t001:** Univariate Cox’s regression analysis of clinicopathological characteristics for overall survival of 485 patients with lung adenocarcinoma (LUAD).

Characteristic	No. (%)	OS	
		CHR (95% CI)	*p*-Value
**Pathology stage**			
I + II	381 (78.6)	1.00	
III + IV	104 (21.4)	2.76 (2.00–3.80)	< 0.001
**pT stage (*n* = 482)**			
T1 + T2	423 (87.2)	1.00	
T3 + T4	62 (12.8)	2.43 (1.65–3.60)	< 0.001
**pN stage (*n* = 481)**			
N0	323 (66.6)	1.00	
N1	89 (18.4)	2.49 (1.73–3.58)	< 0.001
>N2	73 (15.0)	3.31 (2.24–4.88)	< 0.001
**pM stage**			
M0	460 (94.8)	1.00	
M1	25 (5.2)	2.23 (1.31–3.80)	0.003

Abbreviation:.OS, overall survival; CHR, crude hazard ratio.

**Table 2 ijms-21-08833-t002:** Correlation of *FAM83A* expression with clinicopathological characteristics of lung adenocarcinoma patients (LUAD).

Variables	FAM83A (*n* = 511)			
	No. (%)	Mean ± SD	Median	*p*-Value
**Pathology stage**				
I	275 (53.8)	18.05 ± 25.58 ^cd^	10.32	< 0.001 ^a^
II	126 (24.7)	24.28 ± 30.47 ^c^	16.16	
III+ IV	110 (21.5)	41.67 ± 66.33 ^d^	20.54	
**pT stage (*n* = 508)**				
T1	169 (33.1)	15.29 ± 18.70 ^ef^	10.81	< 0.001 ^a^
T2	276 (54.0)	26.81 ± 39.51 ^e^	14.64	
T3 + T4	66 (12.9)	39.07 ± 67.64 ^f^	17.20	
**pN stage (*n* = 507)**				
N0	339 (66.3)	19.33 ± 30.56 ^gh^	11.00	< 0.001 ^a^
N1	95 (18.6)	27.61 ± 34.32 ^g^	16.85	
>N2	77 (15.1)	43.91 ± 67.96 ^h^	21.45	
**pM stage**				
M0	485 (94.9)	23.09 ± 32.62	12.72	0.531 ^b^
M1	26 (5.1)	54.10 ± 105.81	18.65	

^a^*p*-values were estimated by Kruskal-Wallis 1-way ANOVA test. ^b^*p*-value were estimated by Mann-Whitney U test. ^c^*p* = 0.003, ^d^
*p* < 0.001, ^e^
*p* < 0.001, ^f^
*p* < 0.001, ^g^
*p* = 0.001, ^h^
*p* < 0.001.

**Table 3 ijms-21-08833-t003:** Univariate and multivariate Cox’s regression analysis of gene expression for overall survival of 485 patients with lung cancer.

Characteristic	No. (%)	OS			
		CHR (95% CI)	*p*-Value	AHR (95% CI)	*p*-Value
FAM83A	(*n* = 485)				
Low	374 (77.1)	1.00		1.00	
High	111 (22.9)	2.57 (1.87–3.54)	<0.001	2.17 (1.57–3.01)	<0.001

OS, Overall survival; CHR, crude hazard ratio; AHR, adjusted hazard ratio. AHR were adjusted for AJCC pathological stage (II, III and IV vs. I).

**Table 4 ijms-21-08833-t004:** Correlation of miR-1-3p expression with clinicopathological characteristics of 453 lung cancer patients.

Variables	miR-1-3p (*n* = 453)			
	No. (%)	Mean ± SD	Median	*p*-Value
**Pathology stage**				
I	248 (54.7)	13.24 ± 18.01	8.07	0.325 ^a^
II	115 (25.4)	10.18 ± 12.27	5.19	
III	73 (16.1)	10.88 ± 10.13	6.49	
IV	17 (3.8)	12.71 ± 20.41	6.13	
**pT stage**				
T1	157 (34.7)	15.18 ± 16.56 ^e^	10.01	0.002 ^b^
T2	238 (52.5)	10.02 ± 15.63 ^e^	5.95	
T3	43 (9.5)	12.68 ± 13.92	6.55	
T4	15 (3.3)	10.20 ± 7.87	6.61	
**pN stage**				
N0	306 (67.5)	12.53 ± 17.29	6.74	0.772 ^a^
N1	81 (17.9)	11.46 ± 13.12	7.67	
N2	65 (14.3)	10.80 ± 10.48	6.49	
N3	1 (0.3)	3.03 ±	3.03	
**pM stage**				
M0	436 (96.2)	12.04 ± 15.58	6.74	0.863 ^c^
M1	17 (3.8)	12.71 ± 20.41	6.13	

^a^*p*-value were estimated by one-way ANOVA test. ^b^*p*-values were estimated by Kruskal-Wallis 1-way ANOVA test. ^c^*p*-value were estimated by Student’s *t* test, ^e^
*p* < 0.001.

**Table 5 ijms-21-08833-t005:** Univariate and multivariate Cox’s regression analysis of miR-1-3p expression for overall survival of 433 patients with lung cancer.

Characteristic	No. (%)	CHR (95% CI)	*p*-Value	AHR (95% CI)	*p*-Value
miR-1-3p	(*n* = 433)				
Low	326 (75.3)	1.00		1.00	
High	107 (24.7)	0.48 (0.28–0.83)	0.009	0.54 (0.31–0.94)	0.029

Abbreviation: OS, Overall survival; CHR, crude hazard ratio; AHR, adjusted hazard ratio. AHR were adjusted for AJCC pathological stage (II, III and IV vs. I).

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
