# Peer review of "Involvement of MicroRNA-1-FAM83A Axis Dysfunction in the Growth and Motility of Lung Cancer Cells"

_ijms, 2020, doi:10.3390/ijms21228833_

Round 1

Reviewer 1 Report

Authors revisions are done according to comments and al that was unclear was explained.
I believe that manuscript can be accepted.

Author Response

We thank reviewer for suggestions to make this article better.

Reviewer 2 Report

I believe that the manuscript gained significantly from the revision process and I hope the authors share my same feeling. My feedback is positive. I do have still a comment regarding the expression of miR-1 in normal vs. cancer tissue. From the rebuttal, it is my understanding that the expression of miR-1 in normal lung might be very low/not-detectable. MiR-1 is indeed considered to be a "myomiR", being a miRNA expressed specifically in muscle. In figure 6 A and C, I would substitute "normal" with "adjacent normal" or "adj. normal" to avoid misleading the reader. I also believe that it would be important for the authors to clarify this point in the Discussion. I believe they could indicate that although miR-1 expression may be low/not detectable in normal lung tissue, a clear reduction of its expression was observed in the cancer vs. adjacent normal tissue and indeed miR-1 expression in cancer correlated with FAM83A and survival. This might indicate that changes in miR-1 expression are happening in cancer and that a level of its expression is maintained in the adjacent normal tissue, possibly to downregulate FAM83A. Further studies will be needed to determine whether the levels of miR-1 are indeed different in normal vs. "adjacent normal" tissue.

I hope the authors appreciate the suggestion.

Author Response

Based on the reviewers’ comments, we added some sentences in discussion in the revised manuscript. Changes to the manuscript are highlighted in yellow.  

Reviewer#2

I believe that the manuscript gained significantly from the revision process and I hope the authors share my same feeling. My feedback is positive. I do have still a comment regarding the expression of miR-1 in normal vs. cancer tissue. From the rebuttal, it is my understanding that the expression of miR-1 in normal lung might be very low/not-detectable. MiR-1 is indeed considered to be a "myomiR", being a miRNA expressed specifically in muscle. In figure 6 A and C, I would substitute "normal" with "adjacent normal" or "adj. normal" to avoid misleading the reader. I also believe that it would be important for the authors to clarify this point in the Discussion. I believe they could indicate that although miR-1 expression may be low/not detectable in normal lung tissue, a clear reduction of its expression was observed in the cancer vs. adjacent normal tissue and indeed miR-1 expression in cancer correlated with FAM83A and survival. This might indicate that changes in miR-1 expression are happening in cancer and that a level of its expression is maintained in the adjacent normal tissue, possibly to downregulate FAM83A. Further studies will be needed to determine whether the levels of miR-1 are indeed different in normal vs. "adjacent normal" tissue.

I hope the authors appreciate the suggestion.

Response: We thanks reviewer for this suggestion and we have substituted "normal" with "adjacent normal" in figure 6A and 6C. Furthermore, we also added some sentences to discuss it.

In Discussion section, page 26

In this study, we reported for the first time that low miR-1-3p expression may contribute to FAM83A overexpression in human lung cancer. MiR-1-3p was significantly downregulated in lung cancer tissues compared with in adjacent normal tissues. Previous study indicated that miR-1-3p is a conserved miRNA with high expression in the muscle tissues, but low abundant in normal lung tissues. [25] Consistent result also was observed in GTEx Portal (https://www.gtexportal.org/home/gene/MIR1), that miR-1-3p rarely expressed in human normal lung tissues. In present study, miR-1-3p was indicated to have higher expression in the corresponding adjacent normal tissue than LUAD by analyzing TCGA database. These results implied that miR-1-3p expression may be gradually induced when the lung tissue begins to develop precancerous lesions. However, the expression levels of miR-1-3p might be further reduced in a malignancy LUAD. This hypothesis needs more experiments to demonstrate it in future.   

This manuscript is a resubmission of an earlier submission. The following is a list of the peer review reports and author responses from that submission.

Round 1

Reviewer 1 Report

In the present study, Liu et al report for the first time that a miR-1-FAM83A axis is involved in lung cancer growth and that its alteration is associated with poor survival in LuAD patients. The finding are interesting and relevant to the field, suggesting that high expression of FAM83A/low expression of miR-1-3p can be used as biomarker for poor lung cancer prognosis. I have some comments and minor edit suggestions.

Major comments:

1- The authors show that FAM83A is highly expressed in lung cancer, hence they decide to knock it down in lung cancer cells. Before showing the downregulation, the expression at mRNA and protein level of FAM83A should be showed in H1355 and A549 at baseline.

2- The authors use H1355 and A549 cells or most of their studies. However, in figure 3I, as well as for the microarray and for the luciferase assay they only use A549 cells. It is not reported how the choice of using mainly A549 cells was made. Although for microarray and luciferase assay the choice can probably be justified (it needs to be explained in the result section), data on H1355 should be showed in in figure 3I.

3- The paragraph on the microarray is not well connected with the rest of the 'story'. The findings regarding the signaling pathways are not explored further and just listed without any connection with the following paragraph and/or the FAM83A/miR-1-3p axis. The authors should try to make a connection between these sections. Also, in the discussion they mention that the microarray data 'suggest' that microarray approach cannot detect post-translational modification. This is not a surprise and it cannot be considered a conclusion from the data presented, microarrays cannot detect alteration at protein level by definition. Especially because the  microarray was performed with n= 2 instead than (at least) n = 3, the authors need to strengthen the findings, discuss more or consider not including the microarray in the manuscript. It is also peculiar that the paragraph describes data that are completely in the supplementary section. 

4 - MiR-1 is a muscle specific microRNA. It has been indeed reported that its expression is altered in lung cancer and its upregulation can inhibit lung cancer proliferation. I just wonder whether the expression of miR-1 is indeed detected in healthy lung and what is the Ct value at which the authors detect it in the NC.

5- The authors perform the luciferase assay on A549 and report that luciferase activity is downregulated by miR-1. Do they also find a downregulation of FAM83A protein due to miR-1 overexpression? 

Minor comments:

1- In Methods, real-time section, the sequence of primers for miR-1 expression detection should be reported.

2- In Methods, western blotting section, the authors need to indicate the antibodies and the company they were purchased from.

3- In Supplementary Table 1 and 2, please indicate the gene ID together with the gene symbol.

4- In the legend of figures 3,4 and 6, please indicate what NC stays for and specify that it is transfected with a random siRNA sequence. Also, indicate the n for experimental repeats.

Author Response

Reviewer#1

In the present study, Liu et al report for the first time that a miR-1-FAM83A axis is involved in lung cancer growth and that its alteration is associated with poor survival in LuAD patients. The finding are interesting and relevant to the field, suggesting that high expression of FAM83A/low expression of miR-1-3p can be used as biomarker for poor lung cancer prognosis. I have some comments and minor edit suggestions.

Major comments:

1. The authors show that FAM83A is highly expressed in lung cancer, hence they decide to knock it down in lung cancer cells. Before showing the downregulation, the expression at mRNA and protein level of FAM83A should be showed in H1355 and A549 at baseline.

Response: We thank the reviewer for their suggestion. We performed a Western blot assay to examine the expression levels of FAM83A in lung cancer with siRNA transfection. Our data revealed that FAM83A, which was expressed in A549 and H1355 cells, displayed reduced expression after siRNA transfection (Figure 5A and 5B). We did not explore whether the expression levels of FAM83A is higher in human lung cancer cell lines than in normal lung cells because of a lack of normal lung cells in our laboratory.

2. The authors use H1355 and A549 cells or most of their studies. However, in figure 3I, as well as for the microarray and for the luciferase assay they only use A549 cells. It is not reported how the choice of using mainly A549 cells was made. Although for microarray and luciferase assay the choice can probably be justified (it needs to be explained in the result section), data on H1355 should be showed in in figure 3I.

Response: Our results revealed that the biological function of FAM83A knockdown is very similar in A549 and H1355 cells. Therefore, we only analyzed the gene expression profiles of FAM83A knockdown in A549 cells by using a microarray approach. Pathway enrichment data revealed that FAM83A knockdown induced inhibition of lung cancer cell growth and motility might occur through modulation of the choline metabolism signaling pathway. Therefore, we further examined EGFR/MAPK/CHKA signaling activity in lung cancer cells after FAM83A knockdown. Our data indicated that FAM83A knockdown suppressed EGFR/MAPK/CHKA signaling activation in A549 and H1355. The data and a description were included in our revised version.

Abstract section, page 2:

…………………..Our results revealed FAM83A overexpression in lung cancer tissues compared with adjacent normal tissues. Furthermore, high FAM83A expression was closely associated with poor lung cancer survival. Here, through siRNA transfection, we effectively inhibited FAM83A expression in the lung cancer cell lines H1355 and A549. FAM83A knockdown significantly suppressed the proliferation, migration, and invasion ability of these cells. Furthermore, FAM83A knockdown could suppressed EGFR/MAPK/CHKA signaling activation in A549 and H1355. By using a bioinformatics approach, we found that FAM83A overexpression in lung cancer may result from miR-1-3p downregulation. In summary, we identified a novel miR-1-FAM83A axis could partially modulate the EGFR/choline phospholipid metabolism signaling pathway, which suppressed lung cancer growth and motility. Our findings provide new insights for the development of lung cancer therapeutics.

Results section, page 19:

To assess the mechanism of FAM83A knockdown–induced inhibition of growth and invasion ability of lung cancer cells, we performed transcriptome profiling using a microarray approach. The biological function of FAM83A knockdown is very similar in A549 and H1355 cells. Therefore, we only performed gene expression profiles on A549 cells with FAM83A knockdown. From the microarray data, we identified the putative signaling pathways in FAM83A that suppress lung cancer growth and motility.

3. The paragraph on the microarray is not well connected with the rest of the 'story'. The findings regarding the signaling pathways are not explored further and just listed without any connection with the following paragraph and/or the FAM83A/miR-1-3p axis. The authors should try to make a connection between these sections. Also, in the discussion they mention that the microarray data 'suggest' that microarray approach cannot detect post-translational modification. This is not a surprise and it cannot be considered a conclusion from the data presented, microarrays cannot detect alteration at protein level by definition. Especially because the microarray was performed with n= 2 instead than (at least) n = 3, the authors need to strengthen the findings, discuss more or consider not including the microarray in the manuscript. It is also peculiar that the paragraph describes data that are completely in the supplementary section.

Response: We attempted to clarify the relationship between microarray data and FAM83A-related biological function in lung cancer cells. Pathway enrichment analysis revealed that FAM83A is involved in the choline metabolism signaling pathway. Furthermore, choline kinase is overexpressed in lung cancer and has been demonstrated to play a critical role in human cancer onset (PMID: 12176020, 15994937). Therefore, we further examined the effect of choline metabolism activation in lung cancer cells with FAM83A knockdown. Our data indicated that the expression levels of EGFR, MAPK, and CHKA were reduced, and the phosphorylation of EGFR and MAPK was also reduced in lung cancer cells with FAM831 knockdown. These results indicated that FAM83A knockdown induced inhibition of lung cancer cell growth and motility was partially caused by the suppression of the EGFR/MAPK/CHKA signaling activity. We have added these findings in the revised version, deleted inappropriate descriptions, and added a description in the Discussion section.

Results section, page 20:

Pathway enrichment analysis revealed that these 404 genes are critically involved in 16 signaling transduction pathways: Alzheimer disease, thermogenesis, choline metabolism in cancer, renin secretion, Parkinson disease, oxidative phosphorylation, gap junction, glutamatergic synapse, cortisol synthesis and secretion, long-term potentiation, degradation of valine, leucine, and isoleucine, endocytosis, complement and coagulation cascades, propanoate metabolism, legionellosis, and the phospholipase D signaling pathway (Supplementary Figure 3C). We deduced that choline metabolism signaling may be involved in FAM83A knockdown–induced inhibition of growth and motility ability of lung cancer cells. Therefore, we examined the EGFR/MAPK/CHKA signaling activity in A549 and H1355 cells further with FAM83A knockdown. As illustrated in Figures 5A and 5B, the expression levels of EGFR, MAPK, and CHKA and the phosphorylation of EGFR and MAPK were reduced in lung cancer cells with FAM83A knockdown. These results indicated that FAM83A knockdown–induced inhibition of lung cancer cell growth and motility was partially caused by EGFR/MAPK/CHKA signaling activity suppression.

Discussion section, page 27:

FAM83A overexpression promotes cell proliferation and metastasis in lung cancer through the modulation of cancer-related signaling pathways, including PI3K/ATK/Snail, Wnt, Hippo, and MAPK.[26-29] FAM83A knockdown was found to altered several signaling pathways; however, microarray data analysis did not reveal involvement of FAM83A knockdown in PI3K/ATK/Snail or MAPK signaling, suggesting that microarray approaches cannot detect the posttranslational modification such as phosphorylation. Studies have reported that choline kinase is overexpressed in lung cancer and has been demonstrated to play a critical role in the onset of human cancer (PMID: 12176020, 15994937). We revealed that FAM83A knockdown–induced inhibition of lung cancer cell growth and motility was partially caused by the suppression of EGFR/MAPK/CHKA signaling activity. Jiao et al. reported that miR-1-3p expression reduced the phosphorylation of EGFR in lung cancer cells (PMID:29664235). Similar results were observed for neck squamous carcinoma cells by using miR-1 to mimic transfection (PMID: 27169691). Furthermore, in silico analysis revealed that miR-1-3p may be targeted to the 3′UTR of FAM83A. These findings suggest that the miR-1-FAM83A axis is critical in lung cancer cell growth and metastasis.

Figure 5 legend, page 21:

Figure 5. FAM83A Knockdown–Induced Inhibition of Lung Cancer Cell Growth and Motility by Choline Metabolism Signaling. After 5 min of EGF treatment, the expression levels of EGFR, p-EGFR, MAPK, p-MAPK, and CHKA in (A) A549 and (B) H1355 cells with FAM83A knockdown were assessed using Western blot assay. N.C indicates a random siRNA sequence control. All experiments were performed in triplicate, and data were analyzed using Student’s t test. Differences were considered significant when p < 0.05.

4 - MiR-1 is a muscle specific microRNA. It has been indeed reported that its expression is altered in lung cancer and its upregulation can inhibit lung cancer proliferation. I just wonder whether the expression of miR-1 is indeed detected in healthy lung and what is the Ct value at which the authors detect it in the NC.

Response: We attempt to examine the expression levels of miR-1 in normal lung tissues using data obtained from public databases, such as GTEx Portal (https://www.gtexportal.org/home/gene/MIR1). The expression levels of miR-1 are very low in human healthy lung tissues. We did not examine the expression levels of miR-1 in normal lung cancer. We only assessed the expression levels of corresponding adjacent tissues and the effects of FAM83A and miR-1 in lung cancer tissue by analyzing the TCGA database.

5- The authors perform the luciferase assay on A549 and report that luciferase activity is downregulated by miR-1. Do they also find a downregulation of FAM83A protein due to miR-1 overexpression?

Response: Our study revealed that FAM83A expression could be silenced by targeting the 3′UTR of FAM83A using miR-1. We attempted to examine the expression levels of FAM83A in A549 using miR-1 to mimic transfection with the Western blot assay. However, the efficiency of the antibodies in recognizing FAM83A proteins was poor. Our data revealed that FAM83A expression was slightly reduced in A549 and H1355 with miR-1 mimic transfection. This data has been included in our revised version (Figure 7C), with an explanation.

Results section, page 24:

Furthermore, both FAM83A mRNA and protein expression levels in lung cancer cells was considerably reduced after miR-1 mimic transfection (Figures 7A, 7B, and 7C). We cloned the 3′UTR sequence into the pmiR-REPORT Luciferase miRNA Expression Reporter Vector System.

Minor comments:

1. In Methods, real-time section, the sequence of primers for miR-1 expression detection should be reported.

Response: We have added information on the primer sequencing of miR-1.

Methods section, page 8

Stem–loop reverse transcription PCR

According to the manufacturer’s instructions (Invitrogen, Carlsbad, CA, USA) and as we described previously [18], total RNA was reverse transcribed through a stem–loop reverse transcription reaction by using miR-1-3p reverse transcription primers (5′-CTCAACTGGTGTCGTGGAGTCGGCAATTCAGTTGAGATACATAC-3′) and SuperScript III Reverse Transcriptase. Gene expression was detected using an SYBR Green I assay (Applied Biosystems) and the expression level of miR-1-3p was normalized to that of U6 (ΔCt = miR-1-3p Ct-U6 Ct). The sequence of primers was as follows:

miR-1-3p-GSF: 5′- CGGCGGTGGAATGTAAAGAAGT-3′

Universal reverse: 5′- CTGGTGTCGTGGAGTCGGCAATTC-3′

U6-F: 5′-CTCGCTTCGGCAGCACA-3′

U6-R: 5′-AACGCTTCACGAATTTGCGT-3

miR-1-3p overexpression

A549 cells were transfected with 10 nM of miRNA-1-3p mimics (sense: 5′-UGGAAUGUAAAGAAGUAUGUAU-3′; antisense: 5′-ACAUACUUCUUUACAUUCCAUU-3′) or a random siRNA sequence as control (N.C) using Lipofectamine RNAiMAX reagent (13778150, Thermo Fisher Scientific Inc., Waltham, MA, USA).

miRNA target candidates and luciferase reporter assay

Prediction of the miRNA candidates for binding to 3′-UTR of FAM83A mRNA was done using the prediction tool on microRNA.org (Ref). The 3′-UTR sequences and seed region mutant of FAM83A were cloned into a pMIR-REPROT vector (AM5795, Thermo Fisher Scientific). Subsequently, the pMIR-REPROT-FAM83A or pMIR-REPROT-FAM83A(mutant) vector was cotransfected with or without the miR-1-3p expression vector into a 293T cell line using Lipofectamine 2000 (Invitrogen, Thermo Fisher Scientific). After 24 h of transfection, cell lysates were used to measure luciferase activity using the Dual-Glo Luciferase Assay System (Promega Corporation, Madison, WI, USA). The detail information was described in our previous study (PMID: 33109562).

  • In Methods, western blotting section, the authors need to indicate the antibodies and the company they were purchased from.

Response: We have added this information in the revised version.

Methods section , page 8:

Western Blotting

The total cell lysates were extracted with the radioimmunoprecipitation assay buffer (50 mM Tris HCl, pH 8.0, 150 mM NaCl, 1% NP-40, 0.5% deoxycholic acid, 0.1% sodium dodecyl sulfate). Total proteins were separated through electrophoresis in 6%-10% sodium dodecyl sulfate-polyacrylamide gel and transferred onto nitrocellulose filter membranes (Millipore, Billerica, MA, USA). The membranes were then incubated with a blocking buffer for 1 h at room temperature, and incubated with the primary antibodies overnight at 4 °C. For a description of these antibodies, please refer to our previous study.[20] The membranes were then incubated with a horseradish peroxidase–conjugated secondary antibody for 1 h at room temperature to detect the primary antibody. In this study, the primary antibodies were used: CCNB1 (1:1000; 55004-1-AP, Proteintech Group, Inc.), CCND1 (1:200; MA5-16356, Thermo Fisher Scientific Inc.), CDKN1B (p27) (1:500; 25614-1-AP, Proteintech),  FAM83A (1:500, 20618-1-AP, Proteintech), EGFR (1:1000; #4267, Cell Signaling Technology, Inc.), p-EGFR (1:1000; #3777, Cell Signaling Technology, Inc.), MAPK (1:1000; #9107, Cell Signaling Technology, Inc.), p-MAPK (1:1000; #4370, Cell Signaling Technology, Inc.), CHKA (1:1000; #13422, Cell Signaling Technology, Inc.), and ACTB (1:5000; MAB1501, EMD Millipore, Billerica, MA, USA).

2. In Supplementary Table 1 and 2, please indicate the gene ID together with the gene symbol.

Response: We have added gene symbols in Supplementary Tables S1 (page 8).

3. In the legend of figures 3,4 and 6, please indicate what NC stays for and specify that it is transfected with a random siRNA sequence. Also, indicate the n for experimental repeats.

Response: We have clarified the figure legends in the revised version.

In figure 3, 4, 5 and 7, we added these sentences in figure Legends : N.C. indicates a random siRNA sequence control. All experiments were performed in triplicate, and the data were analyzed using Student’s t test. Differences were considered significant when p < 0.05

Reviewer 2 Report

Authors submitted manuscript titled ‘Involvement of MicroRNA-1-FAM83A Axis Dysfunction in the Growth and Motility of Lung Cancer Cells’ to International Journal of Molecular Sciences.

            Authors showed that FAM83A plays a significant oncogenic role in regulating lung cancer cell growth and motility. Additionally, authors reported for the first time that miR-1-3p dysfunction may contribute to FAM83A overexpression and that high FAM83A expression could be a biomarker for poor lung cancer prognosis.

            In the manuscript, authors presented in details about the methods and test used. First, they identified from TCGA database genes whose dysfunction is involved in lung cancer progression.  Afterword’s, authors assessed the relationship between FAM83A expression and postoperative survival of patients with lung cancer. Authors showed that patients who are high expressors of FAM83A have shorter survival compared to low expressors.

            Further, to explore the role of FAM83A in lung cancer, authors assessed its effects on lung cancer cell growth by knocking down its expression in H1355 and A549 cells through transfection of siFAM83A#1 or 214 siFAM83A #2.  FAM83A knockdown substantially inhibited the migration 218 and invasion ability of both the A549 and H1355 cells. Authors also showed that FAM83A knockdown substantially inhibited the migration 218 and invasion ability of both the A549 and H1355 cells

Authors further attempted to identify the miRNAs that regulate FAM83A expression in lung cancer and managed to show that low miR-1-3p expression was significantly correlated with advanced stage and worse overall 262 survival in lung adenocarcinoma. Authors conclude that these findings indicate that miR-1-3p suppresses tumor growth by regulatingthe growth and motility of lung cancer cells. FAM83A is a novel target gene for miR-1-3p in lung cancer, and FAM83A overexpression may result from downregulation of miR-1-3p thus potentially become a novel biomarker.

I believe that manuscript is worth for publication in IJMS after some corrections which are stated below.

  1. Describe in more details in methods section about patients’ characteristics. Which staging system (7 or 8, or other) was used.
  2. Do metastases sites show a difference (brain vs liver vs bones vs other)?
  3. In the results section, manuscript line 145, it is stated that 24 candidates’ genes with down regulation are involved in oncogenesis. However, in Figure 1. It is stated 26 candidates’ genes. Which is correct?
  4. Please describe in more details who you calculated cut-off value for FAM83A expression.

Author Response

Reviewer#2

Authors submitted manuscript titled ‘Involvement of MicroRNA-1-FAM83A Axis Dysfunction in the Growth and Motility of Lung Cancer Cells’ to International Journal of Molecular Sciences.

            Authors showed that FAM83A plays a significant oncogenic role in regulating lung cancer cell growth and motility. Additionally, authors reported for the first time that miR-1-3p dysfunction may contribute to FAM83A overexpression and that high FAM83A expression could be a biomarker for poor lung cancer prognosis.

            In the manuscript, authors presented in details about the methods and test used. First, they identified from TCGA database genes whose dysfunction is involved in lung cancer progression.  Afterword’s, authors assessed the relationship between FAM83A expression and postoperative survival of patients with lung cancer. Authors showed that patients who are high expressors of FAM83A have shorter survival compared to low expressors.

            Further, to explore the role of FAM83A in lung cancer, authors assessed its effects on lung cancer cell growth by knocking down its expression in H1355 and A549 cells through transfection of siFAM83A#1 or 214 siFAM83A #2.  FAM83A knockdown substantially inhibited the migration 218 and invasion ability of both the A549 and H1355 cells. Authors also showed that FAM83A knockdown substantially inhibited the migration 218 and invasion ability of both the A549 and H1355 cells

Authors further attempted to identify the miRNAs that regulate FAM83A expression in lung cancer and managed to show that low miR-1-3p expression was significantly correlated with advanced stage and worse overall 262 survival in lung adenocarcinoma. Authors conclude that these findings indicate that miR-1-3p suppresses tumor growth by regulatingthe growth and motility of lung cancer cells. FAM83A is a novel target gene for miR-1-3p in lung cancer, and FAM83A overexpression may result from downregulation of miR-1-3p thus potentially become a novel biomarker.

I believe that manuscript is worth for publication in IJMS after some corrections which are stated below.

1. Describe in more details in methods section about patients’ characteristics. Which staging system (7 or 8, or other) was used.

Response: Our clinical data were obtained from the TCGA database, and the eighth edition of the TNM staging system was employed.

Methods section, page 5:

Gene Expression Profiles According to Cancer Genome Atlas Data

The Cancer Genome Atlas (TCGA) program collects both cancerous and corresponding normal tissues from patients with LUAD and LUSC. We accessed downloaded TCGA data on RNA sequences in LUAD and LUSC tissues from the Genomic Data Commons Data Portal. The clinical information of patients with LUAD and LUSC was also downloaded. The expression profiles of 515 LUAD and 59 corresponding adjacent normal tissue samples as well as 501 LUSC and 49 corresponding adjacent normal tissue samples were obtained from the TCGA data portal. The transcriptome profiles of 485 patients with LUAD and 494 patients with LUSC were used to perform overall survival analysis using the Kaplan–Meier method. The clinical pathological stages were assessed using the eighth edition of the TNM staging system.

2. Do metastases sites show a difference (brain vs liver vs bones vs other)?

Response: Only 26 patients displayed distance metastasis in LUAD tissue, and only 7 patients displayed distance metastasis in LUSC tissues. Furthermore, the expression levels of FAM83A were not significantly correlated with the pM stage. These results indicated that further analysis of the effects of FAM83A expression on different distance metastasis sites was unnecessary.

3. In the results section, manuscript line 145, it is stated that 24 candidates’ genes with down regulation are involved in oncogenesis. However, in Figure 1. It is stated 26 candidates’ genes. Which is correct?

Response: We apologize for this mistake and we have revised the manuscript.

Results section, page 11:

Compared with those with stage I+II lung cancer, patients with stage III+IV lung cancer demonstrated significantly increased expression of 12 candidate genes but significantly decreased expression of 26 candidate genes (Supplementary Table 1). As Figure 1B shows, FAM83A expression was significantly upregulated in LUAD tissues compared with in adjacent normal tissues.

4. Please describe in more details who you calculated cut-off value for FAM83A expression.

Response: We have included the optimal cutoff values in the manuscript.

Results section, page 15:

Patients with LUAD were separated into two groups representing high and low FAM83A expression on the basis of this cutoff value (30.4). The Kaplan-Meier survival curves showed that compared with low levels, high expression levels of FAM83A were closely associated with shorter survival duration (Figure 2D). Univariate Cox regression analysis revealed that high expression of FAM83A was correlated with poor survival (cHR 2.57, 95% CI 1.87-3.54, P <.001; Table 3). Multivariate Cox regression analysis found high FAM83A expression to be an independent prognostic biomarker for overall survival in LUAD (adjusted hazard ratio [aHR] 2.17, 95% CI1.57-3.01, P <.001; Table 3). Based on the defined cutoff values (19.6), a significant correlation between high FAM83A expression and poor survival in LUSC was demonstrated in both univariate (cHR 1.52, 95% CI 1.14–2.05, P = 0.005)…………………………….